# Preclinical Advances of Therapies for Laminopathies

**DOI:** 10.3390/jcm10214834

**Published:** 2021-10-21

**Authors:** Louise Benarroch, Enzo Cohen, Antonio Atalaia, Rabah Ben Yaou, Gisèle Bonne, Anne T Bertrand

**Affiliations:** Sorbonne Université, Inserm, Institut de Myologie, Centre de Recherche en Myologie, 75013 Paris, France; louise.benarroch@inserm.fr (L.B.); e.cohen@institut-myologie.org (E.C.); a.atalaia@institut-myologie.org (A.A.); r.benyaou@institut-myologie.org (R.B.Y.); g.bonne@institut-myologie.org (G.B.)

**Keywords:** *LMNA*, lamin A/C, progerin, laminopathy, treatment, therapy, HGPS, preclinical models

## Abstract

Laminopathies are a group of rare disorders due to mutation in *LMNA* gene. Depending on the mutation, they may affect striated muscles, adipose tissues, nerves or are multisystemic with various accelerated ageing syndromes. Although the diverse pathomechanisms responsible for laminopathies are not fully understood, several therapeutic approaches have been evaluated in patient cells or animal models, ranging from gene therapies to cell and drug therapies. This review is focused on these therapies with a strong focus on striated muscle laminopathies and premature ageing syndromes.

## 1. Introduction

Type V intermediate filaments, also known as lamins, are the main constituents of the nuclear lamina, a protein meshwork underlining the inner face of the nuclear envelope (NE) and facing chromatin and nucleoplasm. Lamins are divided into two categories: A-type lamins, encoded by the *LMNA* gene, and B-type lamins, encoded by the *LMNB1* and *LMNB2* genes. They display an N-terminal unstructured head domain, a central helical rod domain involved in their assembly into filaments and a globular C-terminal tail that contains a nuclear localization signal and an immunoglobulin-like (IgG-like) fold involved in protein-protein interactions [1].

The *LMNA* gene has 12 exons, among which exon 10 contains an alternative splice site giving rise to two major isoforms: lamin A and C. These two isoforms are identical in their first 566 amino-acids and vary in their C-terminal tail, with six unique carboxyl-terminal amino-acids for lamin C and 98 for lamin A [2]. Unlike lamin C, lamin A is synthesized as a precursor named prelamin A, that contains a C-terminus “CaaX” motif (“C” for cysteine; “a” for aliphatic amino acid; “X” for any amino acid). The C-terminal tail of prelamin A undergoes several post-translational modifications to become mature: (a) farnesylation of the cysteine from the CaaX motif responsible for the anchorage of prelamin A to the NE, (b) the cleavage of the “aaX” motif by zinc metallopeptidase STE24 (ZMPSTE24) and Ras-converting CAAX endopeptidase 1 (RCE1), (c) methylation of the farnesylated cysteine by isoprenylcysteine carboxyl methyltransferase (ICMT), and (d) the cleavage of the last 15 amino-acids including the farnesylated cysteine by ZMPSTE24, releasing mature lamin A from the NE. Like prelamin A, B-type lamins are synthesized as precursors, and while the three first steps of their maturation are similar to lamin A, the last step of maturation, corresponding to the second cleavage by ZMPSTE24, does not occur. Consequently, B-type lamins are suggested to be more closely associated to the NE than lamin A/C [3].

The lamins form parallel dimers that assemble longitudinally in a head-to-tail manner to form a long polar polymer that further associate laterally forming ~3.5 nm thick mature apolar filaments within the ~14 nm thick nuclear lamina under the NE [4,5]. A-type lamins are also found in the nucleoplasm, in a less structured and less complex organization. It has been proposed that nucleoplasmic lamins form dimers or short polymers interacting with intranuclear binding partners [6].

Over 500 mutations, mainly dominant, have been identified throughout the *LMNA* gene and linked to a broad spectrum of diseases called laminopathies. Different groups of diseases have been described based on the main affected tissue: striated (skeletal and cardiac) muscle laminopathies (SML), peripheral neuropathies, familial partial lipodystrophy and multisystemic disorders including premature aging syndromes [6]. In this review, we describe the pathophysiological mechanisms implicated in laminopathies, i.e., the diseases due to *LMNA* gene mutations, with a focus on SML and premature ageing syndromes, and associated preclinical therapies that have been developed over the years.

## 2. Laminopathies’ Clinical Spectrum

### 2.1. The Striated Muscle Laminopathies

SML are defined as a group of diseases generally characterized by dilated cardiomyopathy with conduction and/or rhythm defects (DCM-CD) associated or not with muscular dystrophy of variable type and age of onset. In fact, since the identification of the first laminopathy, the autosomal dominant Emery-Dreifuss muscular dystrophy (EDMD) [7], a range of muscular dystrophies of various clinical severity related to *LMNA* mutations has been described, from the most severe and early onset congenital form, the *LMNA*-related Congenital Muscular Dystrophy (L-CMD [8]) to less severe and almost adult onset form, the autosomal dominant Limb-Girdle Muscular Dystrophy type 1B (LGMD1B [9]). Patients with SML display varying severity of four limbs atrophy and weakness with or without joint contractures and neck/spine rigidity [10]. SML share a common life-threatening cardiac disease characterized by conduction and/or rhythm defects associated with dilated cardiomyopathy resulting in a high frequency of cardiac sudden death and end-stage hearth failure (corresponding to a clinical stage of advanced heart failure with pronounced symptoms at rest and refractory to maximal medical treatment). This cardiac disease can be the only clinical presentation of the disease without any skeletal muscle involvement [11,12].

Both dominant negative effect and haploinsufficiency have been suggested as disease mechanisms [12,13]. A large proportion of *LMNA* mutations causing SML are point mutations and it has been suggested that such mutations have a dominant-negative effect causing disruption of the lamina and compromising nuclear integrity. Indeed, lamin A/C mutants induce mislocalization of lamin A/C interacting proteins, such as lamin B1, lamin-associated protein 2 (LAP2), emerin or nucleoporin 153 (NUP153) [14,15]. In addition, A-type lamin mutants may form altered filaments [16] that aggregate in the nucleoplasm with wild-type (WT) lamin A/C, prelamin A and nuclear factors (such as pRb or SREBP1), which contribute to the pathogenesis of laminopathies [17,18,19,20]. More recently, a study showed that prelamin A was upregulated in hearts of DCM-CD patients and significantly associated with left ventricular (LV) remodeling, suggesting its potential involvement in the progression of cardiac disease [21].

Lamin A/C haploinsufficiency can also cause the nuclear defect underlying the pathogenesis of the disease. Nonsense mutations, out-of-frame insertions/deletions and/or splice site *LMNA* mutations generate truncated proteins that are not detected in patient fibroblasts or in mouse models probably because truncated mRNAs are degraded through nonsense-mediated RNA decay and truncated proteins through proteasome degradation or autophagy [12,13]. The reduced level of lamin A/C has been associated with misshapen nuclei, nuclear envelope disruption, chromatin rearrangement and DNA damage [22,23,24]. Moreover, lamin A/C haploinsufficiency led to early-onset programmed cell death of cardiomyocytes, causing DCM in mice [22,25,26,27]. Interestingly, a genotype-phenotype correlation study performed in 27 patients carrying *LMNA* mutations suggested that late-onset phenotypes were associated preferably with truncating mutations whereas more severe and early-onset phenotypes were associated with dominant-negative non truncating mutations [28]. These findings may help in patient management.

### 2.2. Progeria and Other Premature Aging Syndromes

Another group of laminopathies corresponds to premature ageing syndromes, including those involving children, i.e., restrictive dermopathy (RD) [29] and Hutchinson-Gilford Progeria syndrome (HGPS), reported by Hutchinson and Gilford in the late 1880s [30,31], and those involving adults such as mandibuloacral dysplasia type A (MAD-A) and atypical Werner’s syndrome [32,33]. HGPS, an extremely rare disorder (with a prevalence of approximately 1 in 20 million children), is the far most studied premature ageing syndrome. It is characterized by severe growth retardation, failure to thrive, alopecia, osteoporosis, severe atherosclerosis with cardiovascular decline, abnormal skin pigmentation, lipodystrophy, and joint contractures [34,35]. It is mainly caused by aberrant splicing of the *LMNA* gene resulting from a *de novo* synonymous *LMNA* variation in exon 11. This aberrant splicing induces the deletion of 50 amino acids in prelamin A, including the second cleavage site for ZMPSTE24, hence leading to a truncated lamin A that remains farnesylated and named progerin [36,37]. Accumulation of progerin is toxic for the cell and responsible for structural changes in the nucleus [38].

### 2.3. Lipodystrophies of Dunnigan Type

*LMNA* mutations are the most common gene defect responsible for lipodystrophy syndrome of Dunnigan type (type 2 Familial Partial Lipodystrophy or FPLD2), characterized by lack of adipose tissue in the four limbs and its accumulation within the neck and the face, accompanied by metabolic abnormalities. Symptoms of lipodystrophy may partially overlap with adult progeroid syndromes, underlying the important role of *LMNA* in the development and function of fat-storing adipocytes. Seventy-five percent of *LMNA* mutations that causes FPLD2 are missense mutations encompassing the IgG-like domain [39,40,41]. These mutations have been shown to perturb the interaction of lamin A/C with several partners, including SREBP1, a transcription factor involved in adipocyte differentiation [42].

### 2.4. Neuropathies

Only one homozygous *LMNA* mutation (p.Arg298Cys) has been reported in an axonal form of autosomal recessive Charcot-Marie-Tooth type 2 (CMT2B) peripheral neuropathy [43,44] in families originating from North Africa. Patients had distal axonal sensorimotor neuropathy with a proximal involvement of the lower limb muscles in some cases. A wide range of age of onset, course and severities have been reported suggesting that modifier genes may be involved [45].

## 3. Therapies for Striated Muscle Laminopathies

The use of patient materials and animal models (mouse knock-out (KO) and knock-in (KI) models reproducing human mutations, *C*. *elegans* and drosophila) have greatly helped understanding the pathophysiological mechanisms of laminopathies (Figure 1) [46]. Different strategies have been developed over the years that act either on the primary cause of the diseases or on their consequences (i.e., altered pathways), using gene, cells or drug therapies (Table 1). However, for now, apart from one molecule that is currently under clinical trial (see Section 3.3.4), these therapies are only at the preclinical stage for SML.

### 3.1. Gene and RNA-Based Therapies

#### 3.1.1. Targeting Lamin A/C

Homozygous *Lmna*^Δ*8–11*/Δ*8–11*^ mice, first thought to be a knock-out (KO) model for *Lmna*, display growth retardation, skeletal dystrophy and DCM-CD characterized by left ventricular (LV) dilatation and reduced systolic contraction, and die around 8 weeks of age due to the expression of a truncated lamin A mutant at low level [25,47]. Work performed on genetically modified mice has shown that expression of one isoform only (either lamin A or lamin C) can prevent the onset of deleterious phenotypes of *Lmna*
^Δ*8–11*/Δ*8–11*^ mice [48,49]. In line with this, cardiomyocyte-specific expression of WT-lamin A transgene partially restored cardiac function of these mice. It significantly increased contractility and myocardial performance but had no effect on cardiac dilatation. Improvements of cardiac function have a beneficial effect on lifespan (12% median extension) but are limited by the heterogenic expression of *Lmna* transgene in cardiomyocytes (30 to 40% positive cells) [50].

As overexpression of mutant lamin A/C is often associated with toxicity [51], alternative gene therapy approaches for laminopathies tested the possibility to use exon-skipping strategy to remove the exon bearing the mutation. Scharner et al. evaluated the potential of *LMNA* exon 5 skipping by antisense oligonucleotide (AON) in HeLa cells and WT human dermal fibroblasts but trials in affected cells or animal models are still missing [52].

More recently, our group demonstrated the promising role of spliceosome-mediated RNA trans-splicing (SMaRT) as a therapeutic strategy to replace the mutated pre-mRNA by the corresponding WT transcript using an exogenous RNA called pre-trans-spliced molecules (PTM) [53]. PTM molecules designed to replace exons 1 to 5 of the mutated *Lmna* pre-mRNA, allowing for the targeting of 51% of the described *LMNA* mutations, were tested in vitro and in vivo in the *Lmna*^Δ*K*32/Δ*K*32^ mouse model, a KI mouse reproducing a *LMNA* mutation found in severe EDMD and L-CMD [54]. This strategy rescued part of the nuclear phenotype of *Lmna*^Δ*K*32/Δ*K*32^ mouse myotubes in vitro, however the efficiency of PTM’s adeno-associated virus (AAV) delivery was particularly low, leading to an extremely modest increase in lamin A/C mRNA expression preventing any conclusion regarding the survival analysis in vivo [53]. Despite mixed results, this strategy is a promising tool that could be a potential replacement to classical gene therapy [51]. Finally, Salvarani et al. used CRISPR-Cas9 editing tool to decipher the conduction abnormalities associated with *LMNA*-cardiomyopathies of iPSC-derived cardiomyocytes harboring *LMNA* p.Lys219Thr (*LMNA-K219T)* mutation. They showed that *LMNA-K219T* mutation affects excitability and cardiac impulse propagation by repressing *SCN5A* expression, encoding the sodium channel gene, NA_V_1.5, hallmarks that are restored after CRISPR/Cas9 correction [55].

#### 3.1.2. Targeting Lamin-Associated Proteins

In the nucleus, lamins interact with numerous proteins thus involving them in a wide range of nuclear functions such as cell proliferation, genome organization and DNA repair. Lamina-associated polypeptide 2α (*LAP2α*) is a nucleoplasmic protein that interacts with A-type lamin in the nucleoplasm. Interestingly, mutation in *LAP2α* gene causes autosomal-dominant cardiomyopathy and altered its interaction with A-type lamin [56]. In 2013, Cohen et al. showed that Lap2α and retinoblastoma protein (pRB) signaling were up-regulated in *Lmna*^Δ*8–11*/Δ*8–11*^ mice which could contribute to the phenotype of these mice. Therefore, they generated *Lmna*^Δ*8–11*/Δ*8–11*^/*Lap2α*^−/−^ mice in which the depletion of *Lap2α* increased lifespan and bodyweight but was not sufficient to completely rescue *Lmna*^Δ*8–11*/Δ*8–11*^ mouse phenotype since cardiac defect remained the cause of death of these mice. These results highlight the role of Lap2α/pRB pathway in the deleterious phenotype of these mice [57]. However, Pilat et al. performed a similar study in *Lmna*^Δ*K32*/Δ*K32*^ but did not show any beneficial effect of *Lap2α* depletion on the phenotype of these mice [58].

### 3.2. Cell Therapies

Cellular therapies are also a promising tool in treatment of cardiovascular disease, and notably in heart failure. The functional benefits of these therapies are mainly based on the propriety of implanted cells to release paracrine factors that would activate myocardial repair pathways [59]. In 2013, Catelain et al. compared transplantation efficiency of murine embryonic stem cells (ESC) induced into cardiac lineage, and a murine myoblast cell line (D7LNB1) considered at that time as “gold-standard” for cell-based therapy, into the LV wall of *Lmna^H222P^*^/*H222P*^ mouse, a KI mouse model reproducing a mutation found in EDMD patients and mainly responsible for DCM in homozygous mice [60]. Myoblast engraft had a greater transplantation efficacy and improved cardiac functions (stabilization of LV fractional shortening), whereas ESCs failed to integrate in the myocardium of *Lmna^H222P^*^/*H222P*^ mice [61]. Clearly, more research is needed in order to find the best cell type to use for cell therapy in the heart. Many groups are actively working on multipotent and pluripotent stem cells with promising results [62].

### 3.3. Drug Therapies

Drug therapies are the most advanced therapies for SML. The different molecules tested aimed either at reading through a premature STOP codon or at slowing down the progression of the diseases via modulating altered signaling pathways identified mainly by transcriptomic analyses and RNA sequencing of patient material, KO or KI mouse models.

#### 3.3.1. Molecule Targeting LMNA mRNA

Lee et al. generated human iPSC-derived cardiomyocytes from patients carrying different premature termination codon (PTC) mutations in *LMNA* gene that reproduced the pathological hallmarks of *LMNA*-associated cardiomyopathy. In these models, they tested PTC124, a molecule that induces translational read-through over the PTC to restore the production of the full-length protein and evaluated its potential therapeutic effect. PTC124 treatment showed beneficial effect in only one of the two mutants tested by reducing nuclear blebbing, excitation-contraction coupling and apoptosis [63].

#### 3.3.2. Modulation of Chromatin-Associated Protein Activity

Lamin A/C interacts with chromatin and organizes the genome into large territories called lamin-associated domains (LADs) that influence gene expression in a cell type-specific manner [64]. Therefore, it is not surprising that *LMNA* mutations affect LAD organization and modify gene expression [65,66,67]. Numerous transcriptomic analyses performed on the heart of various animal models have revealed a wide variety of altered signaling pathways, even before the appearance of any pathological features [68,69].

Auguste et al. performed RNA sequencing in a mouse model with a cardiac specific depletion of *Lmna* gene (*Myh6-Cre:Lmna^F^*^/*F*^ mice), before the onset of cardiac dysfunction, identifying over 2300 differentially expressed genes. Among them, *BRD4* (Bromodomain-containing protein 4) gene, a regulator of chromatin-associated protein, was upregulated. Daily treatment of *Myh6-Cre:Lmna^F^*^/*F*^ mice with JQ1, a specific BET bromodomain inhibitor, improved cardiac function, fibrosis, apoptosis and prolonged lifespan. These findings highlight BET bromodomain inhibition as a potential new therapeutic strategy for *LMNA*-associated cardiomyopathy [70].

Similarly, cardiac differentiation defects of ESCs from heterozygous *Lmna^H222P^*^/*+*^ mouse have been correlated to altered expression of genes involved in the epithelial to mesenchymal transition. Analysis of the regulatory regions of genes revealed a decreased H3K4me1 deposit on *Twist* and *Mesp1* that was reversed by inhibiting LSD1, the enzyme responsible for H3K4 demethylation. Treatment restored cardiac differentiation of *Lmna^H222P^*^/+^ ESC, and ameliorated heart formation and function in embryos and post-natal *Lmna^H222P^*^/H222P^ mice [71].

#### 3.3.3. DNA Repair and Oxidative Stress

DNA damage in laminopathies have been associated with increased nuclear envelope rupture, altered Ran-GTP gradient or oxidative stress [72,73,74]. Cells respond to stress by activating redox-sensitive transcription factors (TF) such as pRb, p53 (tumor suppressor) and forkhead box O (FOXO) TF [75]. Transcriptomic and RNA-sequencing analyses performed on mouse embryonic fibroblasts (MEF) or heart tissue from various mouse models (*Lmna*^Δ*8–11*/Δ*8–11*^, *Lmna^H222P^*^/*H222P*^, Tg-*LMNA^D300N^*) all led to the identification of a major up-regulation of p53 [33,68,76,77]. Up-regulation of FOXO, NF-κB or TGF-β were also reported in some models [68,78]. These results were corroborated by a transcriptional analysis of cells from patient with *LMNA*-cardiomyopathy, EDMD and FPLD2 [67,79,80,81]. Modulation of FOXO by shRNA or supplementation with NAD+, with its precursor Nicotinamide Riboside, or with AP endonuclease 1 (APE1] required for base excision repair led to increased DNA repair, and ameliorated altered pathways and mouse survival [68,76,82].

We examined the involvement of oxidative stress in the progression of cardiac disease in *Lmna^H222P^*^/*H222P*^ mice and showed that *LMNA* cardiomyopathy is associated with increased oxidative stress and depletion of glutathione (GSH). Treatment of *Lmna^H222P^*^/*H222P*^ mice with N-acetyl cysteine (NAC), a precursor of GSH, restored the redox homeostasis and delayed the onset of LV dilatation and cardiac dysfunction [77].

#### 3.3.4. Inhibition of MAPK Pathways

Lamin A/C has been shown to play a dynamic role in regulating signal transduction by tethering proteins at the NE. Lamin A/C interacts directly with ERK1/2 (Extracellular signal-regulated kinase 1/2), which highlights a potential role of lamin A/C on the regulation of ERK1/2 signaling pathway [83]. Transcriptomic analyses performed on the hearts of pre-symptomatic *Lmna^H222P^*^/*H222P*^ mice and on explanted hearts of patients with *LMNA*-associated dilated cardiomyopathy showed an increased expression of genes implicated in 3 of the 4 MAPK signaling pathways: ERK1/2, JNK and p38α [69,84]. Inhibition of ERK1/2 was achieved using several MEK1/2 inhibitors (PD098059, Selumetinib, compound 8 allosteric macrocyclic MEK1/2 inhibitor). They all target MEK1/2 kinases responsible for ERK1/2 phosphorylation. Treated *Lmna^H222P^*^/*H222P*^ mice showed a significant slow-down of LV dilatation progression, improved cardiac contractility and functions and increased survival [84,85,86,87,88]. Selumetinib was also shown to have a synergic effect when combined with benazepril, an angiotensin II converting enzyme (ACE) inhibitor, a standard medical therapy in heart failure. Of note, ACE inhibition alone delayed the onset of cardiac disease [89]. Treatment with JNK inhibitor SP600125, or p38α inhibitor ARRY-371797, also slowed down the development of cardiac contractile dysfunction [84,90,91]. The beneficial effects of ARRY-371797 in mice led to the first clinical trial, still on going, on the p38α inhibitor in patients with *LMNA*-associated dilated cardiomyopathy (clinicaltrials.gov #NCT02057341). The results of these studies demonstrate that MAPK activation contributes to the pathogenesis of dilated cardiomyopathy caused by *LMNA* mutation but the mechanism leading to MAPK activation remains unknown.

#### 3.3.5. Inhibition of TGF-β Signaling Pathway

Transcriptome and secretome analyses revealed the hyperactivation of TGF-β signaling in hearts of *Lmna^H222P^*^/*H222P*^ mice, prior to the onset of the cardiac disease and leading to elevated TGF-β2 levels in the majority of the patients (EDMD and LGMD1B and other neuromuscular diseases) and in *Lmna^H222P^*^/*H222P*^ mouse sera [80,92]. TGF-β2 neutralizing antibody avoided activation of fibrogenic markers and myogenesis impairment in vitro [80], while the TGF-β receptor (ALK5] inhibitor SB-431542 reduces fibrosis and improves LV functions in *Lmna^H222P^*^/*H222P*^ mouse hearts, in part via lowering the level of active ERK1/2. These findings highlighted TGF-β as a mediator in the pathogenesis of *Lmna*-associated cardiomyopathy [92].

Inhibition of TGF-β signaling was also tested in another mouse model: the *Lmna*-DCM mice, an inducible and cardiomyocyte-specific model of lamin A/C depletion created by Tan et al. by AAV delivery of shRNA targeting *Lmna* mRNA under cardiac specific promoter in 1.5-week-old mice. These mice exhibit marked fibrosis, cardiac dilation and dysfunction, rescued upon treatment with Yy1 (Ying Yang 1], a transcription factor associated with cell cycle progression. Upregulation of *Yy1* led to the upregulation of *Bmp7* expression and the downregulation of *Ctgf* expression, inhibiting TGF-β signaling pathway [93]. These studies provide several lines of evidence supporting TGF-β signaling as potential targets for DCM-CD and cardiac fibrosis.

#### 3.3.6. Targeting Cytoskeleton Proteins

Recently, it has been reported that ERK1/2 interacts directly with cofilin-1, an actin-depolymerizing factor, that lead to the alteration of the sarcomeric actin polymerization, participating in the development of LV dysfunction in *LMNA*-associated cardiomyopathy and muscle weakness. Inhibition of ERK1/2 using selumetinib or other MEK1/2 inhibitors suppressed cofilin-1 phosphorylation and restored LV functions [78,94].

Microtubule, another cytoskeleton constituent, polymer of tubulin proteins, was shown to be impaired in SML. Impairment of the microtubule network triggered abnormal electrical communication between cardiomyocytes and induced cardiac conduction defects in *Lmna^H222P^*^/*H222P*^, *Lmna^N195K^*^/*N195K*^ and *Lmna*^Δ*8–11*/Δ*8–11*^ mice [95,96,97]. Increased phosphorylation and aberrant localization of Cx43 have been reported, in vivo and in vitro, due to microtubule instability [96,97,98]. Stabilization of microtubules using paclitaxel, a microtubule-stabilization agent commonly used in chemotherapy, improved intraventricular conduction defects in *Lmna^H222P^*^/*H222P*^ mice, demonstrating a novel pathophysiological mechanism based on microtubule network and Cx43 displacement [97].

Disorganized desmin network is also observed in SML, triggering nuclear deformation and contractile dysfunction [25,99]. In *Lmna^H222P^*^/*H222P*^ mice, cardiac-specific expression of αB-crystallin (αBCry), a chaperone protein interacting with desmin to maintain cytoskeletal integrity, has cardioprotective effects by improving desmin network, mitochondrial and nuclear defects and ERK1/2 abnormal activation. Overall, *Lmna^H222P^*^/*H222P*^/*α**BCry^+^*^/−^ mice displayed significantly improved cardiac functions. Interestingly, similar results were observed in desmin-depleted *Lmna^H222P^*^/*H222P*^ mice [100]. Increased of desmin protein levels and disorganization of the desmin network were also rescued in *Lmna*^Δ*8–11*/Δ*8–11*^ mice expressing the cardiomyocyte-specific expression of WT-lamin A transgene [50].

#### 3.3.7. Inhibition of WNT/β-Catenin Signaling

Wnt proteins are secreted cysteine-rich glycoproteins involved in several cellular processes such as proliferation, differentiation, apoptosis and senescence. In the absence of Wnt ligand, β-catenin is phosphorylated by glycogen synthase kinase 3-β (GSK3-β) and degraded by the proteasome. When Wnt binds to its receptors, β-catenin accumulates in the cytosol and translocates to the nucleus where it activates gene expression such as connexin 43 (CX43) [101]. In *Lmna^H222P^*^/*H222P*^ mouse hearts, Wnt, β-catenin and Cx43 expressions are decreased [69,84,102]. The pharmacological activation of WNT/β-catenin signaling using 6-bromoindirubin-3′-oxime (BIO), a GSK3-β inhibitor, restored connexin 43, Wnt-1 and β-catenin expressions and improved cardiac functions of *Lmna^H222P^*^/*H222P*^ mice [102]. Similar results were observed in HL-1 cardiomyocytes transfected with *LMNA* p.Asp243Glyfs*4 mutant, where decreased connexin 43 level was restored by lithium treatment, another well-known GSK3 inhibitor [103].

#### 3.3.8. Activation of Autophagy

The mammalian target of rapamycin (mTOR) pathway plays a key regulatory function in cardiovascular physiology (embryonic development, maintenance of cardiac structure and function) and pathology (cardiac hypertrophy, ischemia). mTOR is an atypical serine/threonine kinase that forms two distinct multiprotein complexes, mTORC1 and mTORC2, to exert specific functions in response to environmental stimuli. mTORC1 plays a central role in protein synthesis, cell growth/proliferation and autophagy while mTORC2 regulates cell survival and polarity [104]. Hyperactivation of mTOR signaling has been reported in mouse models of *LMNA*-associated cardiomyopathy [105,106]. Treatment of 4-week old *Lmna*^Δ*8–11*/Δ*8–11*^ mice with rapamycin, a specific inhibitor of mTORC1, or treatment of 14-week-old *Lmna^H222P^*^/*H222P*^ mice with temsirolimus, a rapamycin analog, showed improvement of cardiac functions [105,106]. Similarly, everolimus treatment, another Rapamycin analog, improves fibroblasts phenotypes of patients carrying various *LMNA* mutations associated with EDMD, HGPS and atypical Werner syndrome [107].

## 4. Therapies for Premature Aging Syndromes

Similarly to SML, different strategies have been developed over the years to understand the pathophysiological mechanisms underlying premature aging syndrome (Figure 2) as well as developing strategies to prevent the progression of disease (Table 2).

### 4.1. Gene and RNA Based Therapies

Gene therapy strategy started in 2005, with the use of morpholino antisense oligonucleotides (MAOs) targeting lamin A cryptic splice site, thus restoring normal nuclear morphology in HGPS fibroblasts [112]. Efficacy of this approach was proven in vivo, and splicing modulation even demonstrated a beneficial upregulation of lamin C transcripts compensating for the absence of lamin A [113,114,115]. Additionally, MAOs’ success extends to other progeroid syndromes, including HGPS-like and MAD-B syndromes [116]. Another strategy relies on the suppression of the specific disease-causing *LMNA* transcript using shRNA, which also showed reduction of abnormal nuclear morphology and cell senescence, and improvement of proliferative potential [117]. RNA interference using microRNAs such as miR-9, specifically targeting lamin A for degradation, exerts a protective effect in HGPS neurons [118,119] reviewed in [120]. CRISPR/Cas for direct genome editing [121,122] could also, after evaluation and minimization of their off-target effects, represent a new potential therapeutic strategy for the clinics [123]. Currently, several groups are working towards the development of efficient tools to deliver such gene therapy, with promising results using AAV or lentiviral vectors [121,124,125]. Lentiviral delivery to induce base editing with adenine base editors in cultured HGPS fibroblasts and mouse models resulted in improved cellular phenotype and rescued vascular pathology in vivo [126,127]. More recently, Mojiri et al. tested the impact of lentiviral delivery of telomerase mRNA (TERT) on senescence in human HGPS iPSC-derived endothelial cells and HGPS mouse model. Both models showed improved phenotypes placing hTERT therapy as a viable option for treating vascular disease in HGPS patients [128].

### 4.2. Drug Therapies

#### 4.2.1. Targeting Post-Translational Processing

Because progerin lacks the target site for ZMPSTE24 endoprotease encoded by the missing exon 11, it remains permanently farnesylated and hence anchored to the inner nuclear membrane. The first drugs tested for treatment aimed at inhibiting protein farnesylation. Farnesyltransferase inhibitors (FTIs), the most commonly used therapeutic agent in the field of HGPS treatment, have shown efficacy against disease phenotype in both HGPS cells and mouse models [129,130,131,132,133,134,135,136,137,138], though mitigated by an alternative post-translational modification of the precursor protein in place of farnesylation: the geranylgeranylation [139]. To overcome this, Varela et al. used a combination of statins and aminobisphosphonates that improved nuclear morphology, lifespan, skeletal properties, and reduced growth retardation and weight loss in the previously mentioned models. Based on a similar approach, Blondel et al., identified mono-aminopyrimidines (mono-APs) as inhibitors of farnesyl pyrophosphate synthase and farnesyltransferase to prevent farnesylation, and rescue progeria cell phenotype [140]. Finally, another way progerin may bind to the lamina is through its carboxy methylation by ICMT. Thus, targeting ICMT represents another option to address progerin post-translational processing, and already showed to be promising in the context of HGPS [141]. Preclinical studies and clinical trials using FTIs (lonafarnib) alone or in combination with statins (Pravastatin) and bisphosphonates (zoledronate) highlighted that triple-combination therapies did not add beneficial effect compared to the single-drug treatment [142,143,144,145,146]. Nevertheless, the combination and cocktail therapies of FTIs with mono-APs and/or ICMT inhibitors may potentially be the right strategy for care improvement. In a continuing effort to find the right treatment for each patient and to make the above mentioned literature data readily available to clinicians, databases such as the treatabolome are being developed, compiling an exhaustive list of existing treatments for laminopathies and other rare disorders [147].

#### 4.2.2. Targeting the Protein

Many of the drugs used in therapies for SML were also tested for HGPS cellular and mice models. It is the case for the mTORC1 inhibitor rapamycin, used to induce autophagy and resulting in restored nuclear morphology, delayed onset of cellular senescence and progerin clearance in HGPS cells [148,149,150]. Rapamycin was also shown to restore peripheral heterochromatin and cell cycle dynamics in cells from MAD patients [151]. Other autophagy inducers such as sulforaphane, a vegetable-derived antioxidant, and temsirolimus both showed comparable beneficial results [152,153].

The combination of FTIs with rapamycin induced correction of aberrant genome organization and reduction of DNA damage [154] while its combination with sulforaphane induced progerin clearance, rescued cellular phenotype, increased ATP level, decreased DNA damage and lowered the number of dysmorphic nuclei, despite an enhanced cytotoxicity [155]. The last FTI-based therapy that has been evaluated clinically in combination with rapamycin (everolimus) is still under clinical trial (clinicaltrials.gov #NCT02579044).

Additionally, all-trans retinoic acid (ATRA), for which the *LMNA* promoter contains response elements, has been shown to induce progerin autophagy in combination with rapamycin in HGPS fibroblasts [156]. Injections of the proteasome inhibitor MG132 also resulted in an autophagy-mediated enhanced progerin turnover in HGPS patient fibroblasts, HGPS patient iPSC-derived mesenchymal stem cells, vascular smooth muscle cells, and *Lmna^G609G^*^/*G609G*^ progeric mouse model [157]. MG132 also downregulates serine and arginine rich splicing factor 1 (SRSF-1) and SRSF-5, two RNA binding proteins favouring *LMNA* aberrant splicing, which could also explain decreased progerin expression and ameliorated nuclear defects [157]. This phenomenon is also observed in presence of Metformin, an antidiabetic drug known to downregulate mTOR signalling and SRSF-1 [158,159]. Lastly, it has been shown that progerin disrupts nuclear lamina in interaction with lamin A/C, and that this specific binding was inhibited by JH1, JH4 and JH13, compounds identified through a chemical library screening. JH4 in particular, alleviates nucleus distortion, senescence-associated β-gal activity, increases H3K9me3 level and proliferation in HGPS patient cells, *Lmna^G609G^*^/*G609G*^ and *Lmna^G609G^*^/*+*^ mice [113]. These effects are even more pronounced in the optimized version of JH4 called progerinin [160].

#### 4.2.3. Targeting Downstream Toxic Effects of Progerin Accumulation

##### Oxidative Stress

Antioxidants such as NAC and methylene blue, Rho-associated protein kinase (ROCK) inhibitors (Y-27632, fasudil) or ataxia-telangiectasia-mutated (ATM) inhibitors (KU-60019), all alleviated mitochondrial dysfunction and improved HGPS phenotype in vitro [161,162,163,164]. Impaired mitochondrial function results in vascular calcification, which was improved in vivo with pyrophosphate treatment [165]. Similarly, olipraz, CPDT, compound AI-1 and TAT-14, small molecules that either activate or stabilize the redox sensor NRF2, significantly reduced oxidative stress, ROS levels and HGPS-associated nuclear defects [166]. MG132, previously shown to reduce progerin expression, also activates NRF2 signalling pathway [167]. Of note, reduction of ROS level was also observed using previously mentioned drugs, namely pravastatin/zolenodrate and metformin [133,159]. An in-human clinical trial (Clinical Trials.gov, NCT00879034) involving 37 HGPS patients followed a previous lonafarnib-only trial [145] and employed a combination of lonafarnib, pravastatin and zoledronic acid. This trial showed only additional bone mineral density benefit beyond the previously demonstrated survival improvement already seen with lonafarnib monotherapy [146].

##### NF-κB Pathway

Based on its link with aging, hyperactivation of NF-κB through the JAK/STAT inflammatory signalling pathway was also explored in search of therapeutic molecules. Hence, sodium salicylate, an inhibitor of ATM, NEMO (NF-κB essential modulator) and baricitinib, an inhibitor of JAK1/2, successfully prevented progeroid features [168,169]. Inflammation could also be alleviated via NF-κB activation using an inhibitor of the reprogramming repressor DOT1L (epz-4777) and MG132, which inhibits the secretion of proinflammatory cytokines [170,171].

##### Other Molecules

Protection and restoration of the nuclear lamina was also achieved in HGPS cells and mouse models after administration of remodelin, a chemical inhibitor of the N-acetyltransferase NAT10 [172,173]. In the same way, cellular senescence was addressed by senolytic drugs like ABT-737 [174] or quercetin and vitamin C [175]. Enhanced cellular proliferation was obtained using S-adenosyl-methionine and spermidine [176,177], and nuclear export balance was restored with leptomycin B, a pharmacological inhibitor of exportin-1, which is overexpressed in HGPS [178]. Improvements in DNA damage repair machinery and epigenetic modifications associated with HGPS was achieved after restoration of vitamin D receptors signalling, using 1α,25-dihydroxy vitamin D3 in HGPS fibroblasts [179]. Other promising treatments such as resveratrol and chloroquine, improved DNA damage response in cellular and mouse models by activation and stabilization of SIRT1 and SIRT6 respectively [180,181,182]. Another study demonstrated that in vivo induction of Oct4, Sox2, Klf4 and c-Myc in *Lmna^G609G^*^/*G609G*^ mice ameliorated age-associated hallmarks [183]. Alternatively, growth hormone treatment (GH and IGF-1), given their impact on aging, also provides beneficial results in progeric conditions [34,184,185].

Dietary supplementation and even fecal transplantation were proposed and already show promising results against HGPS [186,187,188,189,190].

Although the therapeutic effects of numerous compounds have been demonstrated in vitro, some of them still need to be validated in vivo. Yet, altogether, these studies demonstrate that therapeutic benefits can be achieved without targeting progerin itself. Addressing therapies in HGPS associated to progerin accumulation may thus rely on multi-approaches combination.

## 5. Therapies for Other Laminopathies

### 5.1. Lipodystrophies

Dietary modifications and daily physical activity can help improving the metabolic complications of lipodystrophy, as well as insulin sensitizers (such as metformin) and lipid-lowering drugs (statins, fibrates). However, this risk associated with atherosclerotic vascular disease in patients with lipodystrophies promotes the need for novel therapy development and better patient care management. Currently, the most promising treatment for this disease is metreleptin, a recombinant leptin, however it is not widely approved at the present time, and newer leptin analogues are still being developed [197] reviewed in [147].

To allow potential drug screening, Wojtanik et al. [198] developed a mouse model, which highlighted an inability of the adipose tissue to self-renew, unlike the loss of fat suggested in the literature. Preclinical studies suggest the use of PPARγ agonists Thiazolidinediones and adiponectin upregulators as potential therapies, with only modest improvements observed in patients [194,199]. As described in HGPS, autophagy induction, statin and antioxidant treatments represent potential therapies also for FPLD2. Indeed, autophagy modulation mediated by *Itm2a* silencing rescued differentiation of 3T3-L1 mouse preadipocytes through the stabilization of PPARγ proteins [195]; and both Pravastatin and NAC reversed ROS production, inflammatory secretions and DNA damages in vitro [196].

### 5.2. Neuropathies

Similarly to what was done for SML and HGPS, a mouse model homozygous for the *LMNA*-related CMT2 mutation (p.Arg298Cys) has been developed. However, despite abnormalities in peripheral nerves, it did not show any disease phenotype [200], therefore slowing the search for therapeutically active molecules. Advances in treatments for lamin-associated neuropathies will have to rely on other CMT models and therapeutic approaches discussed in [201,202].

## 6. Concluding Remarks

After the first identification of a *LMNA* mutation in 1999, quickly followed by the implication of *LMNA* mutations in other disorders, research has rapidly focused on therapeutic approaches. The identification of numerous altered pathways opened pharmacological possibilities and led to the first clinical trial on HGPS patients in 2007 (#NCT00425607) with mitigated results. Despite huge phenotypical variabilities in laminopathies, similar preclinical approaches have been performed in SML, progeria or lipodystrophies. Hence, drugs used with success in a specific laminopathy might be of interest as well in other laminopathies.

Pharmacological approaches may lead quickly to clinical trials as some drugs are already approved by medical authorities in other diseases. However, considering the numerous altered pathways identified, these approaches are only able to slow down the progression of the diseases as they only tackle one or few of the altered pathways.

As for other genetic disorders, great hope is arising from gene therapy. As researchers continue developing in vivo gene editing and reducing the benefit risk ratio of such strategies, future investigations will determine if genetic correction can supplement drug therapies to tackle both *LMNA* mutations and downstream consequences.

## Figures and Tables

**Figure 1 jcm-10-04834-f001:**
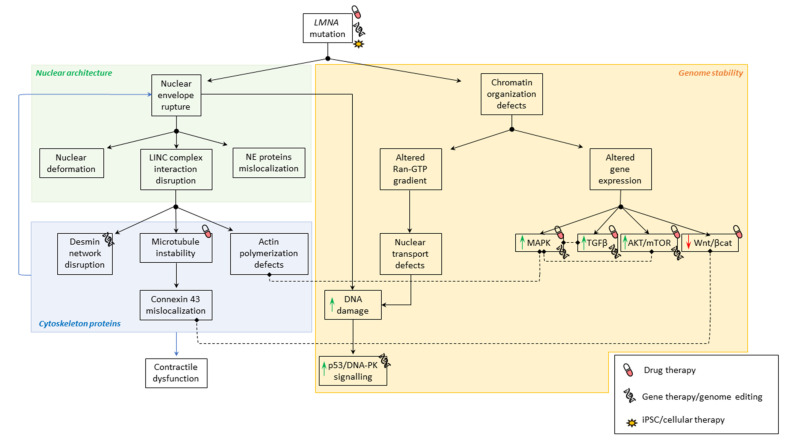
Pathophysiological mechanisms involved in SML. Summary of physiological mechanisms affected in striated muscle laminopathies due to *LMNA* mutations. Black solid arrows indicate the consequence of altered mechanisms. Doted lines indicate correlation between mechanisms.

**Figure 2 jcm-10-04834-f002:**
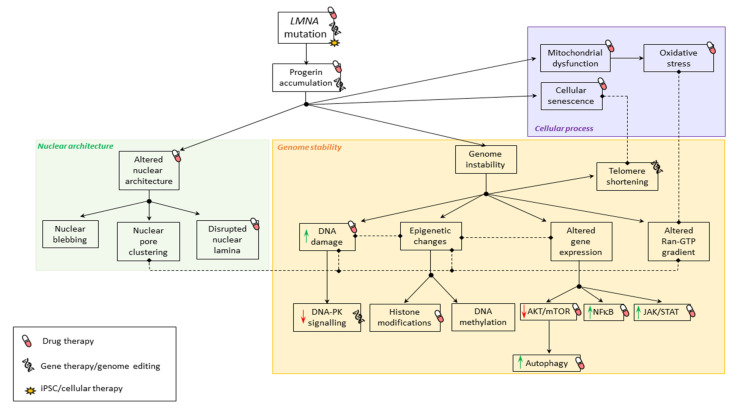
Pathophysiological mechanisms involved in premature aging syndromes. Summary of physiological mechanisms affected in premature aging syndromes due to *LMNA* mutations. Black solid arrows indicate the consequence of altered mechanisms. Doted lines indicate correlation between mechanisms.

**Table 1 jcm-10-04834-t001:** Literature review of preclinical therapeutic strategies in striated muscle laminopathies in vivo and in vitro.

Target	Therapeutical Strategy			Ref.
	Model	Strategy	Benefits	
Gene and RNA-based therapy—*LMNA* gene	·*Lmna*^Δ*8–11*/Δ*8–11*^ mice ·*Lmna*^Δ*8–11*/Δ*8–11*^/*Tg-WT-Lmna* mice	Cardiac-specific WT lamin A transgene	Improvements of cardiac functions, extended lifespan but was limited by the heterogenic expression of *Lmna* transgene in cardiomyocytes	Frock et al., 2012 [50]
	·WT or *Lmna*^Δ*8–11*/Δ*8–11*^ pMEF (murine cells) ·HeLa cells ·Primary WT human dermal fibroblasts	AAV-WT *lamin A* gene or deleted *lamin A* gene AON targeting *LMNA* exon 5	Both lamin A and C lacking exon 5 localized normally at the nuclear envelope and rescued nuclear shape and localization of endogenous lamin B1 and Emerin	Scharner et al., 2015 [52]
	·*Lmna*^Δ*K32*/Δ*K32*^ mice and primary myoblasts	AAV2/9-5′-RNA PTM	Partially rescued nuclear phenotype in vitro but no beneficial effect in vivo	Azibani et al., 2018 [53]
	·*LMNA-*hiPSC-CM	* LMNA^K219T^ * ^ /*WT* ^ -hiPSC-CM CRISPR-Cas 9 corrected hIPSC-CM	Restoration of functional and molecular phenotypes was coupled with decreased binding of lamin A/C to the *SCN5A* promoter	Salvarani et al., 2019 [55]
Lamin-associated proteins	·*Lmna*^Δ*8–11*/Δ*8–11*^ mice ·*Lmna*^Δ*8–11*/Δ*8–11*^/*Lap2a*^−/−^ mice	Genetic depletion of *Lap2a* gene SIS3	Partially rescued *Lmna*^−/−^ mice phenotype (cardiac defect remains the cause of death)	Cohen et al., 2013 [57]
	·*Lmna*^Δ*K32*/Δ*K32*^ mice ·*Lmna*^Δ*K32*/Δ*K32*^/*Lap2a*^−/−^ *mice*	Genetic depletion of *Lap2a* gene	No change on muscle defect in mice	Pilat et al., 2013 [58]
Cell therapy	·*Lmna^H222P^*^/*H222P*^ mice	Transplantation of CGR8 ESC line or D7 Mb cell line in the LV of *Lmna^H222P^*^/*H222P*^ mice	Myoblast engraft improved cardiac functions whereas ESC cells failed to integrate in the myocardium of *Lmna^H222P^*^/*H222P*^ mice	Catelain et al., 2013 [61]
Lamin A/C	·*LMNA*-hiPSC-CM	*LMNA^R225X^*^/*WT*^, *LMNA^Q354X^*^/*WT*^ or *LMNA^T518fs^*^/*WT*^-hiPSC-CM ·PTC124	Beneficial effect on cellular phenotype in one mutant (p.Arg225X)	Lee et al., 2017 [63]
	·*LMNA*-hiPSC-CM	*LMNA^R225X^*^/*WT*^ and *LMNA^FS^*^/*WT*^-hiPSC-CM U0126 and selumetinib	Attenuated or completely abolished the apoptotic effects of field electric stimulation on lamin-deficient cardiomyocytes	Siu et al., 2012 [108]
Chromatin-associated protein	·*Lmna*^Δ*8–11*/Δ*8–11*^ mice	AAV9 expressing a constitutively active form of FOXO TF	AAV9-shRNA-mediated suppression of FOXO TFs partially rescued the molecular (gene expression), biological (apoptosis), and clinical (mortality) phenotypes	Auguste et al., 2018 [68]
	·*Myh6-Cre:Lmna^F^*^/*F*^ mice	JQ1	Improved cardiac function, fibrosis, apoptosis and prolonged lifespan	Auguste et al., 2020 [70]
	·*Lmna^H222P^*^/*+*^-ESC ·*Lmna^H222P^*^/*H222P*^ mice	* Lsd1 * siRNA GSK-LSD1	Rescued the epigenetic landscape of mesodermal cells and contraction of cardiomyocytes. Prevented cardiomyopathy in E13.5 offspring and adults	Guénantin et al., 2021 [71]
DNA repair and oxidative stress	·*Lmna^H222P^*^/*H222P*^ mice	NAC	Restored altered redox homeostasis and delayed cardiac dysfunction	Rodriguez et al., 2018 [77]
	·*Lmna^H222P^*^/*H222P*^ mice ·WT and *Lmna^H222P^*^/*+*^ C2C12 cells ·*LMNA*-hiPSC-CM	* LMNA^R190W^ * ^ /*WT* ^ -hiPSC-CM AAV-cofilin-1 and AAV-cofilin-1-T25A Selumetinib, PD0325901, U0126	Led to F-actin polymerization, prevent cofilin (pThr25] phosphorylation	Chatzifrangkeskou et al., 2018 [78]
	·*Lmna^D300N^ *mice ·*Lmna^D300N^*/*Tp53^F^*^/*F*^ mice	Genetic depletion of *Tp53* gene	Partially rescued apoptosis, proliferation of non-myocyte cells, cardiac functions, and slightly improved survival	Chen et al., 2019 [81]
	·*Lmna^H222P^*^/*H222P*^ mice	NAD+	Improved of the NAD+ cellular content, increased of PARylation of cardiac proteins and improved of LV structure and function	Vignier et al., 2018 [82]
MAPK/ERK signaling pathway	·*Lmna^H222P^*^/*H222P*^ mice	Selumetinib	Partially restored cardiac functions, decreased progression of myocardial fibrosis and extended lifespan	Muchir et al., 2012a [85]
	·*Lmna^H222P^*^/*H222P*^ mice	ARRY-371797	Prevented LV dilatation and deterioration of fractional shortening but did not block the expression of collagen genes involved in cardiac fibrosis	Muchir et al., 2012b [84]
	·*Lmna^H222P^*^/*H222P*^ mice	Selumetinib	Improved skeletal muscle dystrophic pathology and improved function in *Lmna^H222P^*^/*H222P*^ mice	Muchir et al., 2013 [86]
	·*Lmna^H222P^*^/*H222P*^ mice	Allosteric macrocyclic MEK1/2 inhibitor	Improved cardiac functions, has beneficial effects on skeletal muscle structure and pathology and prolongs survival	Wu et al., 2017 [87]
	·*Lmna^H222P^*^/*H222P*^ mice	PD98059	Delayed development of significant cardiomyopathy in *Lmna^H222P^*^/*H222P*^ mice	Muchir et al., 2008 [88]
	·*Lmna^H222P^*^/*H222P*^ mice	Benazepril (ACE), selumetinib	Both ACE inhibition and MEK1/2 inhibition have beneficial effects on LV function in *Lmna^H222P^*^/*H222P*^ mice and both drugs together have a synergistic benefit when initiated after the onset of LV dysfunction	Muchir et al., 2014 [89]
	·*Lmna^H222P^*^/*H222P*^ mice	PD98059, SP600125	Positive effects on cardiac contractility when administered after cardiac dysfunction occurs in *Lmna^H222P^*^/*H222P*^ mice	Wu et al., 2011 [90]
	·*Lmna^H222P^*^/*H222P*^ mice	SP600125	Prevented or delayed the development of significant cardiac contractile dysfunction in *Lmna^H222P^*^/*H222P*^ mice	Wu et al., 2010 [91]
	·*Lmna^H222P^*^/*H222P*^ mice ·*Lmna^H222P^*^/*H222P*^/*Erk1*^−/−^ mice	Genetic depletion of *Erk1* gene Selumetinib	Prevented or delayed the development of LV dilatation and dysfunction, provided a modest albeit not robust survival benefit	Wu et al., 2014 [109]
TGF-β/Smad signaling pathway	·EDMD and LGMD1B patients sera ·Human myoblasts and fibroblasts: controls, EDMD (*LMNA* p.His506Pro), LGMD1B (*LMNA* p.Tyr259Asp) ·*Lmna^H222P^*^/*H222P*^ mice	TGF-β2 neutralizing antibody	Prevented fibrogenic marker activation and myogenesis impairment	Bernasconi et al., 2018 [80]
	·*Lmna^H222P^*^/*H222P*^ mice	SB-431542, PD325901, FG-3019	Inhibition of Tgf-β/Smad signaling pathway suppressed cardiac fibrosis and attenuated cardiac dysfunction	Chatzifrangkeskou et al., 2016 [92]
	·*Lmna* DCM mice	AAV-cTnT-*EGFP*, AAV-cTnT-*Yy1*, AAV-cTnT-*Bmp7*, AAV-cTnT-*Ccdn1* co-expressing shRNA targetting *Bmp7* or *Ccdn1* or *Ctgf*	Overexpression of BPP7 attenuated the suppressive effect of Yy1 on DCM and cardiac fibrosis but not sufficient to suppress DCM and cardiac fibrosis. Upregulation of *BMP7* and *CTGF* silencing significantly suppressed DCM and cardiac fibrosis	Tan et al., 2019 [93]
Cytoskeleton	·*Lmna^H222P^*^/*H222P*^ mice	Paclitaxel (taxol)	Stabilization of microtubule cytoskeleton using improved intraventricular conduction defects	Macquart et al., 2019 [97]
	·*Lmna^H222P^*^/*H222P*^ mice ·*Lmna^H222P^*^/*H222P*^/*αΒCry^+^*^/−^ *mice* ·*Lmna^H222P^*^/*H222P*^/*Des^+^*^/*−*^ *mice*	Genetic overexpression of *αΒCry* gene Genetic depletion of *Des* gene	Improved cardiac functions	Galata et al., 2018 [100]
Wnt/β-Catenin signaling pathway	·*Lmna^H222P^*^/*H222P*^ mice	BIO	Improved cardiac functions	Le Dour et al., 2017 [102]
Autophagy	·*Lmna*^Δ*8–11*/Δ*8–11*^ *mice*	Rapamycin	Rapamycin significantly counteracted mTOR signaling dysfunction, partially restored cardiac functions and ultimately improved survival	Ramos et al., 2012 [105]
	·*Lmna^H222P^*^/*H222P*^ mice	Selumetinib, temsirolimus, chloroquine	Reactivation of autophagy to improved cardiac function	Choi et al., 2012 [106]
	·*LMNA-*mutated fibroblasts: p.Lys35Pro (EDMD); p.Gly608Gly (HGPS); p.Cys588Arg and p.Glu111Lys (atypical HGPS); p.Glu578Val and p.Leu140Arg (atypical WS)	Everolimus	All cell lines showed an increase in proliferative ability and a decreased senescence	Dubose et al., 2018 [107]
	·*Lmna*^Δ*8–11*/Δ*8–11*^ *mice* ·*Lmna*^Δ*8–11*/Δ*8–11*^/*S6K1^+^*^/*−*^ mice	Genetic depletion of *S6k1* gene	Improved muscle skeletal functions and extended lifespan	Liao et al., 2017 [110]
Dusp4	·*Lmna^H222P^*^/*H222P*^ mice ·*Lmna^H222P^*^/*H222P*^/*Dusp4*^−/−^ mice	Genetic depletion of *Dusp4* gene	Increased the medial survival, improved cardiac functions and significantly reduced AKT phosphorylation	Choi et al., 2018 [111]

Therapeutical strategies based on “Gene therapy/Genome editing” are written in blue, “Cellular therapy/iPSC” in green and “Pharmacological treatment” in orange. Des: Desmin; Frameshift; *LMNA*-hIPSC-CM: *LMNA* mutated human-induced pluripotent stem cell-derived cardiomyocytes; NAD+: Nicotinamide Adenine Dinucleotide +; SIS3: Smad3 inhibitors; Tg: Transgene.

**Table 2 jcm-10-04834-t002:** Literature review of preclinical therapeutic strategies in premature aging syndromes and lipodystrophies.

Target	Therapeutical Strategy			Ref. (#)
	Model	Strategy	Benefits	
Premature Aging Syndromes				
Gene and RNA-based therapy—*LMNA* gene	·hHGPS fibroblasts	MAO containing HGPS mutation (*LMNA* p.Gly608Gly)	Restored nuclear morphology	Scaffidi & Misteli 2005 [112]
	·*Lmna^G609G^*^/*G609G*^ and *Lmna^G609G^*^/*+*^ HGPS mice ·hHGPS fibroblasts	JH1, JH4 and JH13 compounds	Efficient blocking of progerin-lamin A/C binding, improving phenotype features in HGPS cells and rescued progeroid features and expand lifespan of HGPS mouse models	Lee SJ et al., 2016b [113]
	·hHGPS fibroblasts	AON targeting *LMNA* exon 11	Reduced alternate splicing in HGPS cells and modestly lowered progerin levels	Fong et al., 2009 [130]
	·*Lmna^G609G^*^/*G609G*^ mice ·*Lmna^G609G^*^/*G609G*^ fibroblasts ·hHGPS fibroblasts	AON targeting *LMNA* splicing site	Decreased progerin levels and rescued nuclear phenotype in vitro; rescued HGPS phenotypes and expanded lifespan in HGPS mice	Osorio et al., 2011 [115]
	·hHGPS fibroblasts and HGPS-like ·*ZMPSTE24*-mutated fibroblasts (MAD-B)	AON targeting *LMNA* exon 10 and 11	Upregulation of lamin C expression and partially restores nuclear morphology and cell senescence	Harhouri et al., 2016 [116]
	·hHGPS fibroblasts	shRNA targeting unspliced G608G mutant pre-mRNA or mature spliced Δ50 *LMNA* mRNA	Restored nuclear morphology, cell senescence and proliferation	Huang et al., 2005 [117]
	·*Lmna^PLAO−5NT^ *mice ·*Lmna^PLAO^*^−*UTR*^ mice	Mutation in miR-9 binding site Prelamin A 3’UTR replaced by lamin C UTR	Protective effect of reduced expression of prelamin A in the brain of HGPS patients	Jung et al., 2014 [118]
	·*Zmpste24*^−/−^ mice ·*Lmna^LAO^*^/*LAO*^ mice ·*Lmna^nHG^*^/*+*^ and *Lmna^nHG^*^/*nHG*^ mice ·forebrain-specific *Dicer*-KO mice ·Neural progenitor cells	miR-9 overexpression	Decreased lamin A and progerin expressions	Jung et al., 2012 [119]
	·*Lmna^G609G^*^/*G609G*^ mice and MEFs ·*LMNA^G608G^*^/*+*^ human fibroblasts	CRISPR-Cas targeting *LMNA* exon 11	Improved nuclear phenotype in vitro and improved HGPS mice phenotype	Santiago-Fernandez et al., 2019 [121]
	·*Lmna^G609G^*^/*G609G*^ mice ·HGPS/*Cas9* mice	CRISPR-Cas targeting *LMNA* exon 11	Improved HGPS phenotype and expand lifespan	Beyret et al., 2019 [122]
	·HGPS-iPSCs	HDAd-based gene correction: *LMNA* c.C1824T	Maintained genetic and epigenetic integrity and reversed HGPS associated phenotypes	Liu et al., 2011 [124]
	·hHGPS fibroblasts	SIRT6 overexpression	Delayed senescence, improved nuclear phenotypes, DNA damages and increased proliferation	Endisha et al.,, 2015 [125]
	·hHGPS fibroblasts ·*Lmna^G609G^*^/*G609G*^ mice	ABE correcting HGPS mutation	Decreased progerin levels and rescued nuclear phenotype in vitro; rescued the vascular pathology and fibrosis, expanded lifespan in HGPS mice	Koblan et al., 2021 [126]
	·hiPSC-EC	ABE correcting HGPS mutation	Decreased progerin expression and rescued nuclear phenotype in vitro	Gete et al., 2021 [127]
	·hHGPS-iPSC-EC ·*Lmna^G609G^*^/*G609G*^ mice	Telomerase mRNA: mTERT, hTERT	Improved replicative capacity; restored endothelial functions and reduced inflammatory cytokines; rescued cellular and nuclear morphology, and normalized transcriptional profile	Mojiri et al., 2021 [128]
	·*Lmna^HG^*^/*LCO*^ mice	”Lamin C only” (LCO) allele	Improved nuclear phenotype in vitro; and improved survival, bone phenotypes	Yang et al., 2008b [191]
	·hHGPS fibroblasts ·*Srsf2^fl^*^/*fl*^ MEFs ·*Lmna^G609G^*^/*G609G*^ mice	AON targeting *LMNA* exon 11	Upregulation of lamin C expression, reduced SRSF2, *Lmna* exon11 decreasing *Prelamin A* transcript expression and reduced aortic pathology in HGPS mice	Lee JM et al., 2016a [192]
*LMNA* post-translational processing	·HeLa, HEK 293, and NIH 3T3 cells ·hHGPS fibroblasts	Mutagenesis on CAAX motifs in WT Lamin A and progerin Lonafarnib (FTI)	Restored nuclear morphology	Capell et al., 2005 [129]
	·*Zmpste24*^−/−^ mice	ABT-100 (FTIs)	Improved mice phenotype and lifespan	Fong et al., 2006 [130]
	·WT and hHGPS fibroblasts	Tg GFP–LaminA-WT, GFP-LaminA-L647R or GFP–progerin PD169541 (FTIs)	Improved nuclear phenotype	Glynn & Glover 2005 [131]
	·MIAMI cells	Tg GFP-progerin and GFP-lamin A FTI-277 (FTIs)	Partially rescued cellular phenotype	Pacheco et al., 2014 [132]
	·WT and hHGPS fibroblasts ·*Zmpste24^−^*^/*−*^ mice and fibroblasts ·*Lmna^G609G^*^/*G609G*^ mice and fibroblasts ·*Lmna^G609G^*^/*+*^ mice	FTI-277 (FTIs)	Restored mitochondrial function	Rivera-Torres et al., 2013 [133]
	·human fibroblasts: RD (*ZMPSTE24* c.1085dupT, hHGPS, atypical progeria (*LMNA* p.Arg644C) ·*Zmpste24^−^*^/*−*^ MEFs	PB-43 (FTIs) BMS-214662 (FTIs)	Improved nuclear phenotype	Toth et al., 2005 [134]
	·*Lmna^HG^*^/*+*^ mice and MEFs	PB-43 (FTIs)	Progerin localized in nucleoplasm and striking improvement in nuclear blebbing	Yang et al., 2005 [135]
	·*Lmna^G609G^*^/*G609G*^ mice	Tipifarnib (FTI)	Prevent both the onset and the late progression of vascular pathology	Capell et al., 2008 [136]
	·*Lmna^HG^*^/*+*^ and *Lmna^HG^*^/*HG*^ mice and MEFs	ABT-100 (FTIs)	Increased adipose tissue mass, improved body weight curves, and improved bone phenotypes	Yang et al., 2006 [138]
	·*Lmna^HG^*^/*+*^ mice	ABT-100 (FTIs)	Improved survival in HGPS mice	Yang et al., 2008a [138]
	·*Zmpste24*^−/−^ mice ·*Lmna*^Δ*K32*/Δ*K32*^ mice ·hHGPS fibroblasts	Statins & aminobisphosphonates	Restored nuclear morphology, improves the aging-like phenotypes (body weight, growth, lipodystrophy, hair loss and bone defects)	Varela et al., 2008 [139]
	·hiPSC-MSC	* LMNA^G608G^ * ^ /*+* ^ hIPSC-MSC Monoaminopyrimidines	Rescued cellular phenotype	Blondel et al., 2016 [140]
	·*Zmpste24^−^*^/*−*^/*Icmt^hm^*^/*hm*^ mice ·*Zmpste24^−^*^/*−*^/*Icmt^hm^*^/*hm*^ fibroblasts	Hypomorphic *Icmt* allele	Improved proliferation, delayed senescence, increased body weight and improved bone phenotypes	Ibrahim et al., 2013 [141]
Progerin	·hHGPS fibroblasts	Rapamycin	Restored nuclear morphology, delayed senescence and enhanced degradation of progerin	Cao et al., 2011 [148]
	·hHGPS fibroblasts	Tg WT-prelamin A or prelamin A-Δ50 Rapamycin	Lower progerin and WT-prelamin A expression, rescued chromatin phenotypes	Cenni et al., 2011 [149]
	·Mouse *Zmpste24*-mutated MDSPC	Rapamycin	Improved myogenic and chrondrogenic differentiation and reduced apoptosis and senescence	Kawakami et al., 2019 [150]
	·hHGPS fibroblasts	Sulphoraphane	Increased proteasome activity and autophagy and rescued HGPS phenotype	Gabriel et al., 2015 [152]
	·hHGPS fibroblasts	Temsirolimus	Improved proliferation and nuclear phenotype and partially ameliorated DNA damage	Gabriel et al., 2016 [153]
	·hHGPS fibroblasts	FTI-277, Pravastatin Zoledronic acid Rapamycin IGF-1 NAC GGTI-2133	FTIs were most effective in restoring genome organization and rapamycin was the most effective in DNA damage repair	Bikkul et al., 2018 [154]
	·hHGPS fibroblasts	Lonafarnib + sulforaphane	Intermittent treatment with lonafarnib followed by sulforaphane rescued HGPS cellular phenotypes	Gabriel et al., 2017 [155]
	·hHGPS fibroblasts	ATRA + rapamycin	Rescued cell dynamics and proliferation	Pellegrini et al., 2015 [156]
	·hHGPS fibroblasts ·*Lmna^G609G^*^/*G609*G^ mice	MG132	Enhanced progerin turnover and impreoved cellular HGPS phenotypes	Harhouri et al., 2017 [157]
	·hiPSC-MSC ·hHGPS fibroblasts ·mouse *Lmna^G609G^*^/*G609G*^ fibroblasts	* LMNA^G608G^ * ^ /*+* ^ -hIPSC-MSC Metformin	Improved nuclear phenotype and premature osteoblastic differentiation of HGPS MSC	Egesipe et al., 2016 [158]
	·hHGPS fibroblasts	Metformin	Restored nuclear phenotype, delayed senescence, activated AMPK phosphorylation and decreased ROS formation	Park & Shin 2017 [159]
	·*Lmna^G609G^*^/*G609G*^ and *Lmna^G609G^*^/*+*^ mice	Progerinin	Extend lifespan, increased body weight, histological and physiological improvements	Kang et al., 2021 [160]
Oxydative stress	·human fibroblasts: RD (*ZMPSTE24* c.1085dupT, HGPS (*LMNA* p.Gly608Gly)	NAC	Reduced the basal levels of double stand break (DSB), eliminated un-repairable ROS-induced DSB and greatly improved cell population-doubling times	Richards et al., 2011 [161]
	·hHGPS fibroblasts	Methylene blue	Rescued mitochondrial functions and nuclear phenotypes, restored genome stability (perinuclear heterochromatin loss, misregulated gene expression)	Xiong et al., 2016 [162]
	·hHGPS fibroblasts	ROCK inhibitors (Y-27632]	Rescued mitochondrial functions, decreased DSB, improved nuclear morphology	Kang et al., 2017 [163]
	·human fibroblasts: HGPS (*LMNA* p.Gly608Gly) and WS (*WRN mutation)*	ATM inhibitors	Restored mitochondrial function, induced metabolic reprogramming, cellular proliferation and ameliorated senescent phenotype	Kuk et al., 2019 [164]
	·*Lmna^G609G^*^/*+*^ mice	Pyrophosphate	Improved vascular calcification	Villa-Bellosta et al., 2013 [165]
	·hHGPS-iPSCs ·hHGPS-iPSC-MSC	Constitutively activated NRF2 Oltipraz	Ameliorates aging defects in vitro by lowering oxidative stress. In MSC, increased expression of NRF2-regulated antioxidants, decreased ROS levels, rescued HGPS nuclear defects and reduced the number of apoptotic and SA β-gal-positive cells	Kubben et al., 2016 [166]
	·hHGPS fibroblasts	ROCK inhibitors (Y-27632]	Restored mitochondrial function, induced metabolic reprogramming, cellular proliferation and ameliorated senescent phenotype	Park et al., 2018 [193]
NF-kB pathway	·hHGPS fibroblasts	Baricitinib	Restored cellular phenotype, delayed senescence and decreased pro-inflammatory markers	Liu et al., 2019 [169]
NAT10	·*LMNA*-depleted U2OS cells (siRNA) ·hHGPS fibroblasts	·Remodelin	Improved nuclear architecture, chromatin organization, and decreased DNA damage	Larrieu et al., 2014 [172]
	·*Lmna^G609G^*^/*G609*G^ mice ·*Lmna^G609G^*^/*G609G*^/*Nat10^+^*^/−^ mice	Remodelin * Nat10 * gene depletion	Increased lifespan, delayed onset of cardiac defects, restored genome stability	Balmus et al., 2018 [173]
Cellular senescence	·*Lmna^G609G^*^/*+*^ mice ·*Prf1*^−/−^ mice ·*Lmna^G609G^*^/*+*^/*Prf1^+^*^/−^ mice	ABT-737	Delayed cellular senescence and increased median survival	Ovadya et al., 2018 [174]
	·WS-hMSCs ·hiPSC-MSC	* LMNA^G608G^ * ^ /*+* ^ -hIPSC-MSC Quercetin	Delayed senescence, restoration of heterochromatin architecture	Geng et al., 2018 [175]
Cellular proliferation	·hHGPS fibroblasts ·*Zmpste24^−^*^/*−*^ mice	SAMe	Improved proliferation and delayed senescence	Mateos et al., 2018 [176]
	·*Lmna*^−/−^ MEFs ·*Zmpste24^−^*^/*−*^ MEFs and mice	Spermidine	Decreased DNA damage and improved progeroid phenotype	Ao et al., 2019 [177]
Nuclear export (Exportin 1]	·hHGPS fibroblasts	Leptomycin B	Improved senescent cellular morphology, genome stability, nuclear morphology	Garcia-Aguirre et al., 2019 [178]
DNA damage	·hHGPS fibroblasts ·*Lmna^G609G^*^/*G609G*^-iMAF	Calcitriol ATRA Remodelin Lonafarnim + rapamycin shRNA-STAT1	Rescued replication defects, repression of STAT1 pathway, improved aging phenotype	Kreienkamp et al., 2016 [179]
	·*Zmpste24^−^*^/*−*^ mice	Resveratrol	improved mice phenotype, bone defects and extend lifespan	Liu et al., 2012 [180]
	·Mouse models: *Zmpste24*^−/−^; A*tm*^−/−^ mice; *Atm*^−/−^/*Sirt6-tg* ·*atm-1* null *C.elegans* ·Cells: HEK293, HepG2, U2OS, HSF ·*Atm*^−/−^, *p53*^−/−^ and *Sirt6*^−/−^ MEFs	Chloroquine	Amelioration of premature aging features and extend lifespan	Qian et al., 2018 [181]
	·hHGPS fibroblasts ·HEK293	Tg: LaminA-WT; LaminA-C661M; LaminA-L647R TSA HDAC inhibitors (MS275]	Restored lamin AC/HDAC2 interaction	Mattioli et al., 2019 [182]
Cellular reprogramming	·*Lmna^G609G^*^/*G609G*^ mice	Cyclic Induction of Oct4, Sox2, Klf4 and c-Myc (doxycycline administration)	Improved age-associated cellular phenotype (DNA damage, cellular senescence, epigenetic modification, nuclear defects), improved mice phenotype and prolong lifespan	Ocampo et al., 2016 [183]
Metabolic homeostasis	·*Zmpste24^−^*^/*−*^ mice	GH IGF-1	Restored somatotroph axis, delayed onset of progeroid features, extend lifespan	Marino et al., 2010 [184]
	·*Lmna^G609G^*^/*G609G*^ mice ·*Zmpste24^−^*^/*−*^ mice	Methionine restriction Cholic acid	Extend lifespan, decreased inflammation and DNA damage, improved metabolic homeostasis	Barcena et al., 2018, 2019a [186,187]
	·*Lmna^G609G^*^/*G609G*^ mice ·*Zmpste24^−^*^/*−*^ mice	Fecal microbiota transplant	Enhanced health span and lifespan, beneficial effect of microbiome	Barcena et al., 2019b [188]
	·*Lmna^G609G^*^/*G609G*^ mice ·*Lmna^LCS^*^/*LCS*^*Tie2Cre^+^*^/*tg*^ and *Lmna^LCS^*^/*LCS*^*SM22αCre^+^*^/*tg*^ mice	Sodium nitrites	Vessel stiffness and inward remodeling were prevented	del Campo et al., 2019 [189]
	·*Lmna^G609G^*^/*G609G*^ mice	High fat diet	Rescued early lethality and ameliorates morbidity	Kreienkamp et al., 2019 [190]
Lipodystrophies				
PPARγ	·CGL-MDMC ·3T3-L1 mouse preadipocytes	*AGPAT1* or *AGPAT2*-siRNA PPARγ agonists Pioglitazone	Partially rescued the adipogenic defect	Subauste et al., 2012 [194]
*Itm2a*	·3T3-L1 mouse preadipocytes: WT and *Lmna-R482W*	* Itm2a-shRNA *	Rescued preadipocytes differentiation	Davies et al., 2017 [195]
Prelamin A farnesylation	·human coronary artery endothelial cells	Tg: WT-prelaminA and R482W-prelaminA Pravastatin NAC	Prevented endothelial dysfunction, with decreased production of NO, decreased endothelial adhesion of peripheral blood mononuclear cells, and delayed cellular senescence	Bidault et al., 2013 [196]

Therapeutical strategies based on “Gene therapy/Genome editing” are written in blue, “Cellular therapy/iPSC” in green and “Pharmacological treatment” in orange. ABE: Adenine Base Editors; ATM: Ataxia telangiectasia mutated; ATRA: All-trans retinoic acid; CGL: Congenital generalized lipodystrophy; GH: Growth hormone; HDAdV: helper-dependent adenoviral vectors; HG: progerin only, hHGPS fibroblasts: human LMNA-mutated (p.Gly608Gly) fibroblasts, hIPSC: human-induced pluripotent stem cell; hIPSC-EC: hIPSC derived endothelial cells; hIPSC-MSC: hIPSC derived mesenchymal stem cells; HSF: Human skin fibroblasts; iMAF: immortalized Adult Fibroblasts; LAO: lamin A only, MDMC: muscle-derived multipotent cells; MDSPC: Muscle derived stem/progenitor cells; MIAMI: Marrow Isolated Adult Multilineage Inducible cells; MSC: Mesenchymal stem cells; RD: Restrictive dermopathy; ROS: Reactive Oxygen Species; SAMe: S-adenosyl-methionine; TSA: Trichostatin A; WS: Werner syndrome.

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
