# Peer review of "Preclinical Advances of Therapies for Laminopathies"

_jcm, 2021, doi:10.3390/jcm10214834_

Round 1

Reviewer 1 Report

Overall, this is an excellent review that summarizes in an elegant way the current knowledge on therapies for laminopathies. This work will significantly contribute to the field. I have only a few comments as followings;

  1. If references are included in the figures or figure legend, it would help to understand.
  2. If tables include vertical lines, it makes us to understand easily.

Author Response

We would like to thank reviewer for its kind comments.

  1. We agree with reviewer that adding references could enhanced understanding. However, considering the number of references that would need to be added, we believe that our figures will lose clarity.
  2. We now provided tables with vertical and horizontal lines.

Reviewer 2 Report

In this manuscript, Benarroch et al review the state of the art of preclinical studies aiming to identify therapies for laminopathies. The authors provide a concise yet exhaustive overview of nuclear lamins and laminopathies before describe in details various approaches targeting various pathways affected by nuclear lamins and laminopathies.

This is a very comprehensive review, well organized, rich in information, and which reads well. The 2 figures, pending they are large enough in final format, are informative and complementary. The authors should be commended on the Table, which is unique reference of its own (remarkable). The focus of this review (therapeutic approaches) is very timely and will stand in the field of lamins and laminopathies. It should be pending very minor issues are addressed.

I noted a few spelling and grammatical errors to be corrected in revision or in the final editorial process.

Table: if the journal does not allow color, I would suggest replacing color with signs (#, §, * etc); same in table legend.

References: these are missing from the PDF I have reviewed, and were somehow not provided despite a request to the editorial office. So I cannot take a stand on their accuracy.

Author Response

We would like to thank reviewer for its kind comments.

  • We have checked our manuscript carefully for spelling and grammatical errors and now provide a corrected version of our manuscript.
  • We did not found information about color usage in table, but are ready to change color for signs if necessary.
  • We are surprised by the fact that references were not present in the pdf version that  reviewers had received. They were present in the pdf version that we downloaded from the journal website. We are sincerely sorry about that.

Reviewer 3 Report

This review is a very valuable ressource for the reserchers of the field to have an overview of the mechanisms involved in  laminopathic cells/ organisms and for the therapeutical strategies tested. 

Few minor comments: 

  1. Nuance the statment B type lamins remain anchored to the NE. (p.1)
  2. p.2 line 68: Maybe explain what is end-stage heart failure.
  3. p.3 mention the name of the neuropathy mutation
  4. p.3 line 131, maybe you can mention at this point other models than mouse and patient cells even if you do not focus on this later. 
  5. p.3 line 134. It is not clear which is the only molecule which is now under clinical trial, at some point, it would be nice to have the authors opinion on what they think has the best chance to be succesfull and few more words/opinion on the future directions on gene editing and gene therapy in the concluding part. 
  6. Figure 1 is very nice and informative ( I just had to mention it), as well as the tables.
  7. For cell therapies,p.5 it would be nice to have one sentence about human perspectives. 
  8. The tables are very informative. Some abbreviations sometimes makes it difficult to follow (ex: Mb engraft or for the benefits we are not sure to which strategy it related to sometimes). In my point of view do not hesitate to write in a clearer way even if it is longer. 

Reviewer 4 Report

Dear authors,

The review is really good 

Author Response

We would like to thank reviewer for its kind comments.